# Possible Reduction of Cardiac Risk after Supplementation with Epigallocatechin Gallate and Increase of Ketone Bodies in the Blood in Patients with Multiple Sclerosis. A Pilot Study

**DOI:** 10.3390/nu12123792

**Published:** 2020-12-10

**Authors:** María Benlloch, María Cuerda-Ballester, Eraci Drehmer, Jose Luis Platero, Sandra Carrera-Juliá, María Mar López-Rodríguez, Jose Joaquin Ceron, Asta Tvarijonaviciute, Marí Ángeles Navarro, Mari Luz Moreno, Jose Enrique de la Rubia Ortí

**Affiliations:** 1Department of Nursing, Catholic University of Valencia San Vicente Mártir, C/Espartero, 7, 46007 Valencia, Spain; maria.benlloch@ucv.es; 2Doctoral Degree School, Catholic University of Valencia San Vicente Mártir, C/Quevedo, 2, 46001 Valencia, Spain; m.cuerda@hotmail.com (M.C.-B.); joseluisplateroarmero@gmail.com (J.L.P.); 3Department of Basic Sciences, Catholic University of Valencia San Vicente Mártir, C/Ramiro de Maeztu, 14, 46900 Torrente, Valencia, Spain; eraci.drehmer@ucv.es (E.D.); angeles.navarro@ucv.es (M.Á.N.); 4Department of Nutrition and Dietetics, Catholic University of Valencia San Vicente Mártir, C/Quevedo, 2, 46001 Valencia, Spain; sandra.carrera@ucv.es; 5Department of Nursing, Physiotherapy and Medicine, University of Almería, Carretera Sacramento, C/San Urbano, s/n, La Cañada, 04120 Almería, Spain; mlr295@ual.es; 6Interdisciplinary Laboratory of Clinical Analysis, Campus of Excellence Mare Nostrum, University of Murcia, 30100 Murcia, Spain; jjceron@um.es (J.J.C.); asta@um.es (A.T.)

**Keywords:** multiple sclerosis, cardiac risk, epigallocatechin gallate, ketone bodies

## Abstract

Multiple sclerosis (MS) is a neurodegenerative disease that causes anthropometric changes characterised by functional disability, increase in fat mass, and decrease in lean mass. All these variables are related to a greater cardiac risk. The polyphenol epigallocatechin gallate (EGCG) and an increase in ketone bodies in the blood have been shown to have beneficial effects on anthropometric and biochemical variables related to cardiovascular activity. The aim of this study was to analyse the impact of the intervention with EGCG and ketone bodies on cardiac risk in MS patients. A population of 51 MS patients were randomly assigned to a control group and an intervention group (daily dose of 800 mg of EGCG and 60 mL of coconut oil). Both groups followed an isocaloric diet for 4 months. Levels of beta-hydroxybutyrate (BHB), albumin, paraoxonase 1 (PON1) and C-reactive protein (CRP) were measured in serum before and after the intervention, as well as determining functional ability, waist circumference, waist-to-hip ratio (WHR), waist-to-height ratio (WHtR), fat percentage and muscle percentage. After 4 months, in the intervention group there was a significant increase in BHB, PON1 and albumin, while CRP did not vary; a significant decrease in cardiac risk associated with a significant decline in WHR; as well as a significant increase in muscle percentage. By contrast, these changes were not observed in the control group. Finally, results from analysis of variance (ANOVA) revealed a significant time–condition interaction effect, observing that WHtR and fat mass decreased in the intervention group, while they increased in the control group.

## 1. Introduction

Multiple sclerosis (MS) is a chronic and degenerative disease of the central nervous system (CNS) of an inflammatory nature and characterised by neuronal demyelination and reactive astrocyte scar formation [1]. Clinically speaking, patients with MS have functional disabilities [2] that are directly related to a progressive loss in lean mass associated at the same time with an increase in fat mass [3]. These anthropometric changes are observed through an increase in body mass index (BMI) and, especially, waist circumference [4] where fat accumulates around the abdomen [5]. A rise in body fat is positively correlated with concentrations of C-reactive protein (CRP) [6], which is linked to the disease’s progression [7]. In addition, metabolic alterations alongside fat accumulation, especially in the abdominal area, increase oxidative stress [8]. This can be related to a decrease of paraoxonase 1 (PON1) in the blood, an enzyme that inhibits low density lipoproteins (LDL) oxidation which hydrolyses the oxidised lipids formed in LDL and high density lipoproteins (HDL), increases the hydrolysis of lactones, aryl esters and organophosphates [9], thus avoiding cytokines being produced [10] and suppressing the expression of the cytokine-induced endothelial cell adhesion molecule (CAM) and the inhibition of the expression of E-selectin induced by IL-1b [11]. In addition, low activity of PON1 in serum IS linked to the development of neurodegenerative diseases [12], including MS [13].

The anthropometric changes that appear throughout the disease are related to cardiovascular risk, the prevalence of which is higher for MS patients than the general population [14]. On the one hand, a decline in skeletal muscle mass leads to a rise in the specific lipid levels related to hyperlipidemia and the development of cardiometabolic syndromes [15]. On the other hand, an increase in body fat, associated with inflammation and oxidation, multiplies the risk of developing cardiovascular diseases [16], especially when body fat is located in the abdominal area [17]. In addition, progressive loss of functionality in MS patients (established with the Expanded Disability Status Scale (EDSS)) linked to anthropometric changes [3] is also correlated with specific comorbidities, including ischemic heart disease [18]. This cardiac risk is biochemically related to low activity of PON1 [19] that is also related to the pathogenesis of the disease, due to the association between lipoproteins and cholesterol metabolism, and the progression of MS [20]. Its activity is also decreased in the case of relapses [21]. In this same line, CRP is increased, especially in relapses of MS patients when compared to the levels in healthy people [22], and is positively correlated with adiposity, suggesting a greater risk of metabolic and cardiovascular diseases [14]. The majority of cardiovascular diseases begin and develop with an endothelial dysfunction, as well as an increase in inflammatory state and oxidative stress [23]. Serum albumin, the most abundant plasma protein, has many physiological properties, including anti-inflammatory, antioxidant [24,25], and antiplatelet aggregation activity [26]. As a result, low levels of albumin in serum are related to different types of cardiovascular disease, such as coronary artery disease, heart failure, atrial fibrillation and strokes [27]. In addition, significant changes in albumin concentration in the blood in MS patients are detected in the short-term after improvement in circulatory parameters [28]. Thus, hypoalbuminaemia can act as a risk factor for cardiovascular diseases, mainly due to exacerbation of inflammation, oxidative stress and platelet aggregation [27].

Taking these precedents into account, therapeutic alternatives should be considered in order to improve the progression of the disease, therefore decreasing cardiac risk associated with inflammation and oxidation as a result of an increase in body fat. Kim et al. proved that a ketogenic diet administered in MS mice models slowed down the progression of the disease, improving motor disability and hippocampal atrophy [29]. Furthermore, ketone bodies obtained after hepatic beta oxidation have been shown to decrease inflammatory markers, including high-sensitivity CRP, indicating at the same time cardiovascular risk improvement [30]. In terms of the nutrients capable of providing higher rates of ketone bodies in the blood, those with high levels of medium-chain triglycerides (MCTs) stand out. In this sense, coconut oil is possibly the food with the highest amount of MCTs as it has high percentages of medium-chain fatty acids (MCFAs), such as caprylic acid, capric acid and lauric acid [31].

Epigallocatechin gallate (EGCG), the main polyphenol found in green tea, reduces the severity of the disease in animal models of autoimmune encephalomyelitis (EAE), as brain inflammation and demyelination damage decrease [32]. This is evidenced by a reduction in encephalitogenic T cell responses and a lower expression of inflammatory cytokines and chemokines [33]. In this sense, Wu et al., showed that treating rat animal models that had induced cerebral ischemic process with 10 mg/kg orally for 7 days demonstrated a significant improvement in memory deterioration induced by the ischemic process. In addition, levels of glutation (GSH) and superoxide dismutase (SOD) increased, and malondialdehyde (MDA) concentrations decreased significantly, both in the cerebral cortex and hippocampus. It was established that bioavailability achieved in these amounts is sufficient for activity at a central level [34].

In terms of its cardioprotective activity, this polyphenol also has been described to improve many cardiovascular risk markers. In particular, it reduced lipids in the blood on a circulatory level, improved ischemia-reperfusion injury in cardiac myoblast cells, decreased myocardial oxidative stress, improved the endothelial function of blood vessels, attenuated inflammation and protected the function of cardiomyocytes in the myocardial tissue [35]. In addition, a decrease in diastolic blood pressure was observed in obese men after receiving 800 mg of EGCG on a daily basis, therefore reducing cardiac risk [36], as well as a significant decrease in LDL in postmenopausal women after administering 800 mg also on a daily basis [37].

Therefore, the objective of this study is to analyse the impact of supplementation with EGCG and coconut oil as a source of ketone bodies in the blood, on cardiac risk in a population of patients with MS by examining anthropometric variables and biomarkers of cardiovascular risk.

## 2. Materials and Methods

An exploratory/pilot study was conducted by means of a clinical trial. The clinical trial ID for this study is NCT03740295.

### 2.1. Subjects

The population sample was obtained from the main state-wide MS associations who were previously informed about the study. Sixty-seven people volunteered to take part in the study. Eligibility criteria included: patients over 18 years of age diagnosed with MS at least 6 months previously and under treatment with glatiramer acetate and interferon beta. In addition, the possibility of moving without the need for a wheelchair or walker was required to be part of this study. On the other hand, the exclusion criteria applied to the volunteers included: pregnant or breastfeeding women, patients with tracheotomy, stoma or with short bowel syndrome, patients with dementia, those evidencing alcohol or drug abuse, patients with myocardial infarction, heart failure, cardiac dysrhythmia, symptoms of angina or other heart conditions, patients with kidney conditions with creatinine levels two times higher than normal markers, patients with elevated liver markers three times higher than normal or with chronic liver disease, patients with metabolic or endocrine diseases such as hyperthyroidism or diabetes, patients with acromegaly, patients with polycystic ovary syndrome or MS patients who were included in other research with experimental drugs or treatment.

### 2.2. Procedure

Participants received information on the study including the defined objectives and the tests and analyses to be carried out, as well as signing an informed consent. Before the intervention, participants registered their solid and liquid intake from the last 7 days. In order to facilitate this registration, patients were provided with information about the weight of each portion of the different types of food [38] and the most common household measurements were provided, such as a portion, a spoonful, a glass, a slice, a plate or a cup. According to all the collected information and after a personal interview on food habits, an individual isocaloric diet was designed taking into account each patient’s pathophysiology.

Diets were designed using the “Easydiet Programa de Gestión de la Consulta^®^” software (Spanish Academy of Nutrition and Dietetics, Pamplona, Spain) where the anthropometric characteristics of each participant and their pathologies were introduced to establish the caloric, macronutrient and micronutrient needs. They were also provided with instructions to not change the prescribed diet for each case (depending on whether they were in the control group or intervention group), including how to cook the food, portion sizes, the products to be bought at the supermarket, as well as to take the capsules on a daily basis at the scheduled times over the 4-month duration of the intervention. Weekly telephone calls were made by team members to each and every patient on Monday mornings in order to verify whether they were complying with the treatment. These calls were made throughout the whole duration of the intervention on a weekly basis. In addition, the subjects had an appointment with the team every 15 days to interview them and verify whether they were following the diet. They were asked questions (see Appendix A) to understand the level in which they followed the diet. Thus, it was verified that they did not eat foods not included in the prescribed diet, such as those with a lipid profile that may affect the study (mainly butter and goat milk with high levels of medium-chain fatty acids) or that have compounds similar to those of the intervention in their composition (such as tea, coffee, red fruit, rich in polyphenols).

They were also asked about any doubts they had or incidences regarding the diet in order to ensure the caloric intake was followed, as well as whether they had come across any issues with the capsules (such as an intolerance or side effects). No general issues or problems with the diet or capsules were registered.

### 2.3. Intervention

Once the selection criteria had been applied, a final sample of 51 MS patients was obtained and participants were randomly assigned to the intervention and control group. Randomisation without stratification was performed by drawing sealed, opaque envelopes previously arranged in a computer-generated random order. Once the population had been divided into the two groups, the study began on 4 October 2018 and all functional tests were conducted, alongside anthropometric measurements and obtaining blood samples from all participants, which were repeated on 7 February 2019 when the study came to an end. These measurements are indicated in the measurements section. For the 4-month duration of the study, the intervention group followed an isocaloric diet which was individually given and explained. This diet was adapted to the individual characteristics of each patient and divided into 5 meals a day: breakfast, mid-morning snack, lunch, afternoon snack and dinner. It was enriched with 60 mL of extra virgin coconut oil divided into 2 equal intakes (30 mL for breakfast and 30 mL for lunch, for which they were provided with a calibrated syringe, administering the content directly into the mouth), representing: 91.84% saturated fatty acids, 6.23% monounsaturated fatty acids and 1.93% polyunsaturated fatty acids, which were included in the isocaloric diet and adapted to an adult’s nutritional requirements; and supplemented with 800 mg of EGCG (Manufacturer Taiyo green power co. Ltd. CN; batch number 702131,708161) administered in two capsules of 400 mg to be taken twice a day (one capsule in the morning and another in the afternoon), the quantity of which responds to the pharmacokinetic calculations of Feng WY. (2006) [39].

On the other hand, the control group followed the same isocaloric diet as the intervention group for the same 4 months, except for coconut oil. In this group, the diet was also given and explained individually. Furthermore, they were administered placebo tablets (opaque capsules containing microcrystalline cellulose, matching in size and colour). Both groups followed the same instructions. The basal diet for all participants included the following percentage distribution of the 3 main macronutrients with respect to the total caloric value: 20% proteins, 40% carbohydrates and 40% Mediterranean lipids that were not rich in medium-chain fatty acids. This ensured that a state of nutritional ketosis did not occur in the control group, as the percentages and the amounts of macronutrients were not characteristic of a ketogenic diet [40], and nor did the composition of lipids make up the said 40%. However, it was foreseeable to achieve ketosis in the intervention group, as the highest percentage of 40% of lipids was from coconut oil rich in medium-chain fatty acids. In terms of catechins, foods rich in polyphenols, especially tea and coffee, were avoided when designing the basal diet for both groups. This diet was characterised for being balanced, varied and with sufficient calories, by providing adequate food proportions divided into 5 daily intakes. Overall, recruitment/enrolment was performed in a total period of six months (Figure 1).

### 2.4. Measurements

The following measurements were taken before and after the 4-month intervention, in the same conditions and at the same time. In the specific case of the scales, they were carried out by the same neurologist assigned to each patient before the study.

The measurements were taken by an ISAK (International Society for the Advancement of Kinanthropometry) level 3 certified anthropometrist in line with the protocol established by the society [41]. The validated anthropometric material used was: a portable clinical scale, SECA model, with a 150–200 kg capacity and 100 g precision; height rod, SECA model, 220 Hamburg, Germany, with a 0.1 cm precision; metal, inextensible and narrow anthropometric tape, model Lufkin W606PM, with 0.2 mm precision; a mechanical skin fold caliper, model Holtain LTD, Crymych, UK, with a 0.2 mm precision and measurement range from 0 to 48 mm; a bicondylar pachymeter to measure the diameter of small bones, model Holtain, with 1 mm precision and measuring range from 0 to 140 mm; and a dermographic pencil to mark anatomical points. The variables that were measured were: body weight, waist and hip circumference, and tricep, subscapular, supraspinal and abdominal folds. Measurements were taken twice, with a third measurement made in the event that the difference between the first two measurements was greater than 5% for the folds and 1% for the other measurements.

The waist-to-hip ratio (WHR), an anthropometric measurement to measure intra-abdominal fat levels, was calculated using the ratio of the waist circumference to that of the hip [42]. Cardiovascular risk obtained from the WHR variable, and based on the classification offered by Yusuf et al. [43], was rated as high (≥1 in men; 0.85 in women), moderate (0.96–1.0 in men; 0.81–0.85 in women), and low risk (≤0.95 in men; 0.80 in women). The waist-to-height ratio (WHtR) was obtained by calculating the quotient between waist and height measurements [44]. The Faulkner equation was used to calculate lean mass percentage [45]. Bone weight was calculated with the Rocha formula [46] and the Matiegka formula [47] was used to calculate muscle weight with which the percentage of muscle mass was obtained.

Functional ability was also measured with the EDSS [48]. The scale is an ordinal scale based on a neurological examination of the eight functional systems (pyramidal, cerebellar, brainstem, mental, sensory, visual, bowel and bladder), alongside assessing walking capacity, which, as a result, provides a disability index between 0 and 10, 0 being understood as having normal health and 10 death by MS.

Blood tests were carried out in the peripheral vein (antecubital vein) at 11 a.m. on an empty stomach. The blood samples were collected in BD Vacutainer Plus serum blood collection tubes (ref. 367815). Once the test was finished, the samples were left at room temperature for 30 min to coagulate. The coagulated part was separated by centrifuging the samples at 2012 g for 10 min in a refrigerated centrifuge (Thermo Scientific Sorvall ST, San Diego, CA, USA, 40R centrifuge). Once centrifuged, the supernatant liquid (blood serum) was transferred to 0.5 mL aliquots, which were then frozen and stored at −80°C. Finally, the concentration of CRP and albumin were measured by commercial reagents (Beckman Coulter, OSR6147 in the case of CRP and OSR6102 in the case of albumin). Both assays had an imprecision lower than 10% and were linear after serial dilutions. The BHB concentrations were measured with a commercial kit (Randox Laboratories, Crumlin, UK, RB1007) and PON1 activity by using 4-Nitrophenyl acetate (Sigma Chemical, St. Louis, MI, USA) using a previously described assay [49]. All measurements were made in an automated clinical biochemistry analyser (Olympus A 400, Tokyo, Japan).

### 2.5. Ethical Concerns

The study was conducted in accordance with the Declaration of Helsinki [50], prior approval of the protocol by the Human Research Committee of the Experimental Research Ethics Committee of the University of Valencia (procedure number H1512345043343). In addition, patients included in the study signed a consent form after being informed on the procedures and nature of the study.

### 2.6. Statistical Analysis

A statistical analysis was carried out with the SPSS v.23 (IBM Corporation, Armonk, NY, USA) tool. The first step estimated the distribution of the variables investigated through statistical methods in order to assess normality, including the Kolmogorov–Smirnov Test. This analysis showed the non-normal distribution of all the scale variables studied. In addition, the Mann–Whitney U test was used to assess the inter-group and pre-post differences, respectively. Categorical data were analysed with a chi-square test. The analyses also included a two-way (repeated measures; pre-test to post-test) by 2 (between subjects; groups) analysis of variance (ANOVA) model. A *p*-value below 0.05 was considered significant. Data are presented as mean ± standard deviation, or the number of patients and percentage.

## 3. Results

Sociodemographic characteristics of the 51 MS patients with a mean weight of 69.48 kg, divided into an intervention and control group, are shown in Table 1. There were no significant differences between both groups regarding any of the variables analysed in the study, including the categorical variables of gender or MS type.

After the intervention period, some significant differences between both groups were observed. The intervention group showed a significantly lower figure than the control group for WHR. By contrast, the treatment group revealed significantly higher means for muscle mass, albumin and BHB (Table 2). Furthermore, after the 4-month intervention, no significant differences were observed in the control group in terms of cardiac risk. Nonetheless, significant changes were observed in the intervention group regarding the distribution of people in different cardiac risk groups (high, moderate, low). The number of high-risk patients decreased and patients with a moderate or low risk increased (Table 2, Figure 2).

Finally, the results from ANOVA revealed a significant time–condition interaction effect. Therefore, we found that WHtR [F(1.31) = 12,752; *p* = 0.001] and fat mass [F(1.48) = 21,275; *p* = 0.000] of participants in the intervention group decreased from baseline, whereas these variables increased in the control group. In addition, muscle mass [F(1.49) = 7975; *p* = 0.007], albumin [F(1.43) = 8222; *p* = 0.006] and PON1 [F(1.41) = 4501; *p* = 0.040] of participants in the intervention group increased while they decreased in the control group (Table 3).

## 4. Discussion

One of the main problems that MS patients have is a high level of cardiac risk, mainly due to fat accumulation in the abdomen [42,51] and muscle loss [52]. These anthropometric changes are related to cardiac comorbidities [18].

With the aim of improving this cardiac risk, an isocaloric Mediterranean diet was administered and supplemented with high quantities of EGCG and coconut oil, a source of MCT, which increase the levels of ketone bodies in the blood after hepatic metabolism [53]. Thus, a significant increase of BHB in the blood was found in the intervention group. In addition, there was a significant improvement in cardiac risk, as percentages of patients with moderate or lower risk increased. This suggests that this kind of diet could allow the probability of coronary disease to be reduced [54]. The possible role of EGCG in terms of this cardiac improvement, as this polyphenol protects against lipid peroxidation [55], alongside its anti-inflammatory and antioxidant activity, makes administering it to be associated with a lower risk of cardiovascular disease [56]. As previously mentioned, lowering cardiac risk is related to reducing abdominal fat. EGCG has important anti-obesity properties by decreasing adipocyte proliferation, lipogenesis and the production of inflammatory adipokines [57], especially in the abdomen [58], improving the parameters we have measured [59]. Therefore, ketone bodies, in equally isoenergetic conditions, improve basal metabolic rate [60] and lower appetite while increasing the satiating effect. However, these effects are moderate in humans. As a result, weight is lost based on fat loss [61].

Cardiac risk does not only depend on high fat percentages, but also on related muscle loss [62,63]. Accordingly, administering EGCG shows protective and repairing effects on skeletal muscle [64] and, at the same time, improvements have been detected in muscles after following a ketogenic diet [65]. Our results are in agreement with these findings as our treatment achieves an increase in lean mass only in the intervention group.

Effectiveness of the intervention on an anthropometric level seems to be specifically confirmed by fat percentage, WHtR and muscle percentage variables, after applying the ANOVA test that associates both the effect of the duration of the intervention with the treatment itself.

This cardiac risk decrease associated with anthropometric improvements has also been evidenced at a biochemical level. Abdominal fat accumulation directly contributes to the chronic inflammatory state of the disease [5], increasing the concentration of proinflammatory molecules related to oxidation state and cardiac risk. In particular, our study measured albumin, PON1 and CRP before and after the intervention. The albumin has been negatively correlated with the development of MS, even having a more pronounced relation as the disease progresses [66]. In our case, a significant increase only in the treated patients was observed, which showed that our intervention with EGCG and coconut oil increased the concentration of this protein. This effect is beneficial for the heart, as low concentrations of albumin are associated with coronary artery disease or heart failure [67], and hypoalbuminaemia is considered as a cardiovascular risk factor [27]. Moreover, an increase in the levels of albumin could strengthen the effects of EGCG in our treatment, as it has been proved for albumin to bind to EGCG by stabilising this polyphenol and enhancing its antioxidant power [68,69].

We also obtained an equally significant increase in the activity of PON1, in line with what has been published by other authors who, on the one hand, saw a significant increase in the activity of paraoxonase when administering polyphenol resveratrol in mice with atherosclerosis (ApoE-deficient mice) [70], and, on the other hand, administering catechins in human beings significantly increased PON1 in serum, causing a decrease in proinflammatory cytokines [71]. In rats, when following a diet enriched with coconut oil, there was a significant increase in PON1 activity, unlike what happened when they followed a diet enriched with other oils, such as copra oil, olive oil and sunflower oil [72].

Finally, CRP, which is related to the development of MS [14], did not decrease after the intervention. It could be expected that the administration of coconut oil would increase CRP, as has been previously described [73]. However after the intervention of our study, the values did not increase, but they remained with similar levels, which could be due to the compensatory effect of EGCG, whose activity decreases the secretion of CRP [74,75].

All of the positive biochemical changes obtained only in the intervention group are also observed after applying the ANOVA test for albumin and PON1 (whose activity even worsens in the control group), which seems to confirm treatment effectiveness.

Despite these promising results, our study does have some limitations. We believe that measuring the variables should be monitored throughout the whole study, and not only before and after the intervention, thereby trying to delve into the activity at a metabolic level of the treatment. It would also be necessary to assess the evolution in the values of other biochemical markers, especially the different types of lipoprotein that are related to the protection or risk of developing cardiovascular diseases. Furthermore, despite there being no differences in the average weight of both groups (intervention and control), the doses could not be adjusted individually according to weight in the study. Finally, due to the exploratory and pilot nature of this study, more stringently designed intervention trials are required in the future.

## 5. Conclusions

The administration of of the antioxidant EGCG in MS patients produces an increase in ketone bodies in the blood and decreases cardiac risk in MS patients. This decrease in cardiac risk is determined as a result of the improvement of two different variables: anthropometric (such as a reduction in WHR and an increase in muscle percentage) and serum analytes (with an increase in PON1 and albumin in the blood). Therefore, despite the fact that these results should be further studied in depth in future studies, this intervention seems to have potential as a therapeutic alternative to improve cardiac risk in these patients.

## Figures and Tables

**Figure 1 nutrients-12-03792-f001:**
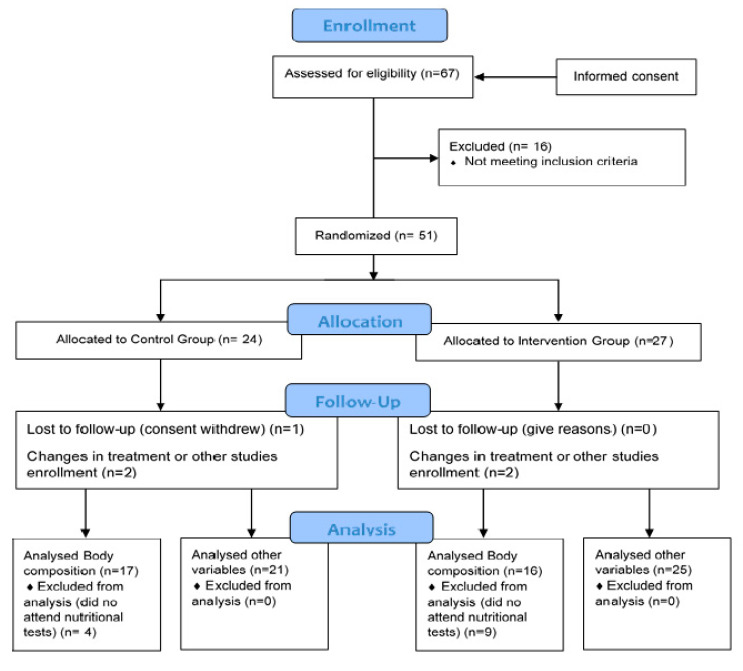
Consort flow diagram.

**Figure 2 nutrients-12-03792-f002:**
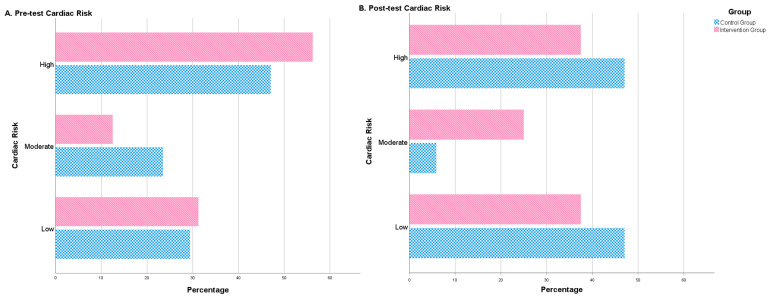
Cardiac risk (percentage of patients) before treatment (pre-test cardiac risk) and after treatment (post-test cardiac risk). Number of patients (in percentage) with low, moderate or high cardiac risk before and after the intervention. The figure shows the differences between the control group (blue) and the intervention group (red).

**Table 1 nutrients-12-03792-t001:** Sociodemographic and clinical characteristics of the population of the study. Comparison between different groups at baseline.

		**Control Group** **N = 17**		**Intervention Group** **N = 16**			
		**Freq**	**%**	**Freq**	**%**	**Chi2**	***p***
Sex	Women	14	58.3%	22	81.5%	3.279	0.070
	Men	10	41.7%	5	18.5%		
MS type	Relapsing-Remitting	17	70.8%	20	74.1%	0.067	0.796
	Secondary-Progressive	7	29.2%	7	25.9%		
Cardiac risk	Low	5	29.4%	5	31.3%	0.696	0.706
	Moderate	4	23.5%	2	12.5%		
	High	8	47.1%	9	56.3%		
		**Control group** **N = 21**		**Intervention group** **N = 25**			
		**Mean**	**SD**	**Mean**	**SD**	**Z**	***p***
Age (years)		49.83	12.42	44.56	11.27	−1.558	0.119
EDSS		3.80	2.00	3.37	2.03	−0.780	0.435
Weight		70.44	18,13	68,63	13,56	−0.245	0.806
BMI		25.72	6.01	25.92	5.29	−0.142	0.887
PON1(UI/L)		2.88	0.77	2.67	0.62	−0.832	0.405
WHR		0.95	0.08	0.89	0.10	−1.625	0.104
WHtR		0.60	0.08	0.57	0.08	−0.721	0.471
Fat mass (%)		18.85	5.00	19.53	3.78	−0.764	0.445
Muscle mass (%)		38.38	4.15	39.39	2.88	−0.547	0.584
Albumin (g/dL)		4.66	0.41	4.69	0.29	−0.414	0.679
CRP (mg/L)		5.03	4.39	3.59	2.08	−0.799	0.424
BHB (Mmol/L)		0.05	0.02	0.06	0.04	−0.932	0.351

MS: Multiple Sclerosis; EDSS: Expanded Disability Status Scale; BMI: Body Mass Index; PON1: Paraoxonase 1; WHR: waist-to-hip ratio; WHtR waist-to-height ratio; CRP: C-reactive protein; BHB: Beta-Hydroxybutyrate; SD: standard deviation; Chi2: Chi square test; Z: Mann–Whitney U test.

**Table 2 nutrients-12-03792-t002:** Comparison between different groups post-test.

		**Control Group** **N = 17**		**Intervention Group** **N = 16**			
		**Freq**	**%**	**Freq**	**%**	**Chi2**	***p***
Cardiac risk	Low	8	47.1%	6	37.5%	2.343	0.310
	Moderate	1	5.9%	4	25.0%		
	High	8	47.1%	6	37.5%		
		**Control group** **N = 21**		**Intervention group** **N = 25**			
		**Mean**	**SD**	**Mean**	**SD**	**Z**	***p***
BMI		25.36	5.85	25.16	4.94	−0.068	0.946
PON1(UI/L)		2.49	0.79	2.97	0.65	−0.369	0.001
WHR		0.94	0.08	0.87	0.09	−1.969	0.049
WHtR		0.60	0.09	0.55	0.07	−1.515	0.130
Fat mass (%)		19.01	5.26	17.74	3.32	−0.701	0.483
Muscle mass (%)		38.01	4.02	41.10	2.81	−2.115	0.023
Albumin (g/dL)		4.55	0.44	4.83	0.19	−2.316	0.021
CRP (mg/L)		4.72	3.55	3.50	2.02	−1.671	0.095
BHB (Mmol/L)		0.04	0.04	0.10	0.10	−2.655	0.008

BMI: Body Mass Index; PON1: Paraoxonase 1; WHR: waist-to-hip ratio; WHtR waist-to-height ratio; CRP: C-reactive protein; BHB: Beta-Hydroxybutyrate; SD: standard deviation; Chi2: Chi square test; Z: Mann–Whitney U test.

**Table 3 nutrients-12-03792-t003:** Comparison of change scores after completion of intervention versus control.

	Control Group N = 21		Intervention Group N = 25			
	Mean *	SD	Mean *	SD	F	*p*
BMI	−0.5182	0.89418	−0.7629	1.03918	0.782	0.381
PON1 Pre-Post (UI/L)	−0.0047	0.40128	0.2427	0.35522	4.501	0.040
WHR Pre-Post	−0.0029	0.03016	−0.0156	0.02421	1.761	0.194
WHtR Pre-Post	0.0006	0.02045	−0.0238	0.01857	12.752	0.001
Fat mass Pre-Post (%)	0.2152	1.54251	−1.7870	1.51905	21.275	0.000
Muscle mass Pre-Post (%)	−0.5958	1.50057	0.8307	2.02952	7.975	0.007
Albumin Pre-Post (g/dL)	−0.1353	0.36632	0.1435	0.28599	8.222	0.006
CRP Pre-Post (mg/L)	−0.3230	5.72099	−0.0889	1.85873	0.040	0.843
BHB Pre-Post (Mmol/L)	−0.0024	0.04309	0.0431	0.09899	3.167	0.042

BMI: Body Mass Index; PON1: Paraoxonase 1; WHR: waist-to-hip ratio; WHtR waist-to-height ratio; CRP: C-reactive protein; BHB: Beta-Hydroxybutyrate; SD: standard deviation; Chi2: Chi square test; F: analysis of variance (ANOVA) test; * Mean change from pre-test to post-test.

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
