# Peer review of "Possible Reduction of Cardiac Risk after Supplementation with Epigallocatechin Gallate and Increase of Ketone Bodies in the Blood in Patients with Multiple Sclerosis. A Pilot Study"

_nutrients, 2020, doi:10.3390/nu12123792_

Round 1
Reviewer 1 Report
Possible Reduction of Cardiac Risk after Supplementation with EGCG and Increase of Ketone Bodies in the Blood in Patients with Multiple Sclerosis. A pilot study.
The work is very interesting and shows new possibilities of therapeutic application of EGCG with coconut oil.
The title is accordance with the content of the manuscript. The obtained results have been correctly interpreted and their value is increased by the use of the ANOVA test.
This is a preliminary study, so the number of people in the study is adequate, and to draw far-reaching conclusions more patients with MS should be tested.
Did the study take into account the patient's weight or the dose of EGCG was constant, which could significantly affect the obtained results.
Appropriate procedures and tests were applied in the research and therefore, in my opinion, the work may be published in the journal.
The manuscript is well written, the study was well designed, the methodology is appropriate and the results are well presented, but conclusion section is too short. Please rewrite the conclusions and underline your results and the importance of your results.
The authors rightly note the limitations of their manuscript. Particular parameters should be determined at specific intervals, not only at the beginning and end of the test. However, this is a pilot study and as such is sufficient.
Please check English and grammar errors.
Reviewer 2 Report
EGCGEGCGEGCG
The Authors analyzed the effect of EGCG administration on cardiac risk in MS patients. The work is interesting and present the impact of supplementation with EGCG.
I would recommend the explanation of EGCG in title.
Reviewer 3 Report
This study’s aim was to improve biomarkers related to cardiovascular risk and inflammation in patients with multiple sclerosis via consumption of ECGC (800mg) and coconut oil (60ml) for 4 months. This treatment group was compared to a group of MS patients that consumed isocaloric diets with similar macronutrient ratios.
Overall, this study was well-designed and produced interesting results. However, there are some issues with the rationale and design of the intervention itself that should be addressed further in the manuscript. For example, coconut oil is a good source of medium chain triglycerides, but these only make up roughly 60% of the oil content. The rest of the oil is saturated long chain fatty acids such as palmitic acid. These oils have been proposed to contribute to cardiovascular risk, for instance increasing LDL. The authors should comment on the controversial nature of coconut oil as an oil with cardiovascular benefits, and should also explain why they did not examine cholesterol and lipids related to cardiovascular risk such as LDL or or triglycerides. Furthermore, although this diet was not intended for weight loss, it is important to understand what weight change occurred during the intervention that may have contributed to the changes in body composition.
Detailed Comments:
Abstract:
No results on differences between groups were reported in the abstract. Since this is a RCT it would be helpful to indicate the presence/absence of group differences.
Introduction:
Line 91: the authors state that palmitic acid, stearic acid, myristic acid and oleic acid and MCFA, which is incorrect.
Line 95: ECGC effects are described only in animal studies, please include examples of cardiovascular biomarker effects in humans, with doses included.
Methods:
Line 130: please comment further on how dietary intake was collected, was it via a validated questionnaire?
Line 150: 60 ml of coconut oil daily is a large amount of oil to incorporate into a diet and has been known to cause gastrointestinal issues, did the subjects gradually increase their intake to help minimize GI issues?
Line 150: What was the average deviation from the calorie budget in both groups? Also, it seems likely that the amount of fat was increased for the control group, how was this accounted for in terms of dietary incorporation (e.g. were subjects asked to add olive oil to their food?)
Line 165: explain the rationale for the doses of coconut oil and ECGC chosen
Line 166: The syringe method of taking the oil seems challenging for compliance, were the subjects allowed to incorporate the oil into their food?
General Methods:
How were subjects randomized?
Please incorporate the ranges of WHR for each cardiac risk category
It’s not clear what was expected in measuring ketone bodies after an overnight fast, since any effect of the coconut oil on blood ketones would’ve been minimal after a 12 hour fast. More useful would’ve been to measure ketones after 30 ml ingestion of the oil, to assess how ‘ketogenic’ this dose is in these types of subjects
Results:
Table 1: please report the average BMI of the groups
Table 2: please report the average change in weight in both groups
Table 3 and Figure 2 show the same information. Please select one for the main article, the other can be included as supplementary material
Discussion:
Line 289: 289: CO is not a source of ketone bodies, it is a source of MCT (about 60%)
Line 300: important to note these effects of MCT consumption (change in metabolic rate, appetite) are very small as reported in humans
Line 333: why look at CRP if it was known to be increased by CO? Were other cardiovascular biomarkers (eg homocysteine) not considered?
Lastly, spelling and grammar need to be thoroughly checked
